# Sustained Cognitive Improvement in Patients over 65 Two Years after Cochlear Implantation

**DOI:** 10.3390/brainsci13121673

**Published:** 2023-12-03

**Authors:** Sophia Marie Häußler, Elisabeth Stankow, Steffen Knopke, Agnieszka J. Szczepek, Heidi Olze

**Affiliations:** 1Department of Otorhinolaryngology, Head and Neck Surgery, Charité–Universitätsmedizin Berlin, Corporate Member of Freie Universität and Berlin Humboldt Universität zu Berlin, Charitéplatz 1, 10117 Berlin, Germany; s.haeussler@uke.de (S.M.H.); elisabeth.stankow@charite.de (E.S.); steffen.knopke@charite.de (S.K.); agnes.szczepek@charite.de (A.J.S.); 2Department of Otorhinolaryngology, University Medical Center Hamburg-Eppendorf, Martinistr. 52, 20246 Hamburg, Germany

**Keywords:** cognition, cochlear implantation, stress, depressive symptoms, speech perception, working memory, processing speed

## Abstract

This study aimed to evaluate the long-term benefits of cochlear implantation (CI) on cognitive performance, speech perception, and psychological status in post-lingually deafened patients older than 65 (*n* = 33). Patients were consecutively enrolled in this prospective study and assessed before, one year after, and two years after CI for speech perception, depressive symptoms, perceived stress, and working memory and processing speed. The Wechsler Adult Intelligence Scale (WAIS) was used for the latter. Thirty-three patients (fourteen men and nineteen women) were included. The scores indicating “hearing in quiet” and “hearing with background noise” improved significantly one year after CI and remained so two years after CI. The sound localization scores improved two years after CI. The depressive symptoms and perceived stress scores were low at the study’s onset and remained unchanged. Working memory improved significantly two years after CI, while processing speed improved significantly one year after CI and was maintained after that. The improvement in working memory and processing speed two years after CI suggests there is a sustained positive effect of auditory rehabilitation with CI on cognitive abilities.

## 1. Introduction

According to the World Health Organization (WHO), more than 5% (430 million people) of the global population currently suffers from disabling hearing loss and requires hearing rehabilitation. By the year 2050, it is expected that more than 2.5 billion people will have some degree of hearing loss, and 700 million will have severe hearing loss [1]. The causative factors contributing to hearing loss include, among others, occupational [2] or during-leisure [3] noise exposure, the use of ototoxic medications [4,5,6], mutations in genes essential for auditory function [7], compression of auditory tissues by tumors (vestibular schwannomas or meningiomas) [8,9], or the effects of cardiovascular diseases [10]. However, one of the most common causes of hearing loss is the aging process [11,12].

The aging process is associated not only with hearing loss but also cognitive decline. Several studies focusing on dementia and hearing impairment provided evidence suggesting that the degree of hearing impairment is related to cognitive decline [13,14,15]. Livingston and colleagues proposed that hearing loss is the most important, modifiable risk factor for age-related dementia among middle-aged persons between 45 and 65 [16,17] and that its treatment could reduce the risk of dementia by 8.2% [17].

Hearing rehabilitation with hearing aids for patients with moderate-to-severe hearing loss was associated with improved cognition [18,19] and suggested to counteract cognitive decline [20,21]. In particular, older candidates with severe-to-profound hearing loss appear to benefit from hearing rehabilitation via CI. A recent systematic review and meta-analysis by Yeo and colleagues demonstrated a decrease of 19% in cognitive decline among hearing-impaired users of hearing aids and cochlear implants [22]. Moreover, in the same work, the authors observed a 3% improvement in the cognitive test scores of this population.

Several theories exist about the relationship between cognitive impairment and hearing loss [23]. The first one, called the “cognitive load hypothesis”, was introduced by John Sweller in the context of educational psychology [24]. It states that an individual requires a certain amount of cognitive effort to perform a task. Learning may be impaired if the learning task is too demanding for the individual’s cognitive capacity and, therefore, working memory. Applying this hypothesis to hearing-impaired patients, it was proposed that they must make an additional listening effort to understand speech and hence have less remaining cognitive capacity for other tasks. Another theory is the “common cause hypothesis”, which states that cognitive and hearing impairments result from neurodegeneration in the aging brain and neuronal pathways [25]. The “cascade hypothesis” suggests that age-related hearing loss (ARHL) leads to changes in brain volume and may cause brain atrophy [26,27]. This may also lead to impaired verbal communication and social isolation-risk factors for cognitive decline [28]. Lastly, the “overdiagnosis hypothesis” suggests that the scores for neuropsychological tests may be lower among people with hearing loss due to hearing impairment. Therefore, the level of cognitive impairment may be overestimated [29].

Cognitive decline can be measured with different instruments. One widely used test is the Mini-Mental State Examination (MMSE), a questionnaire for assessing the degree of cognitive decline. MMSE was also used in a long-term study by Amieva et al. [30], who investigated cognitive differences in older adults with and without hearing loss. Amieva’s study provided evidence that patients with hearing loss who received hearing rehabilitation with hearing aids had less cognitive decline than those who did not receive hearing rehabilitation. Another critical screening instrument for dementia and cognitive decline is the Montreal Cognitive Assessment test (MoCA), which is considered a sensitive tool for detecting mild cognitive impairment and is available in a version for people with hearing impairment [31]. Other useful tools for testing cognition in hearing-impaired and especially CI patients were introduced by Claes et al. [32], with the Repeatable Battery for the Assessment of the Neuropsychological Status for Hearing Impaired Individuals (RBANS-H), and Völter et al. [33], with their computer-based neurocognitive assessment battery for the elderly. An additional instrument measuring the cognitive skills is the Wechsler Adult Intelligence Scale-Fourth Edition (WAIS-IV). The WAIS-IV provides an age-appropriate measure of an individual’s intelligence and cognitive abilities and is one of the most widely used intelligence tests for adults and adolescents worldwide. It was first introduced by David Wechsler in 1955 and is based on the Wechsler–Bellevue Intelligence Scale [34]. In addition to age-specific testing, another critical advantage of the WAIS-IV over screening tests for cognitive impairment and dementia, such as the Mini-Mental State Examination (MMSE) and the Montreal Cognitive Assessment Test (MoCA), is the qualitative and quantitative measurement of the WMI and PSI. The MMSE is widely used as a screening tool for dementia, whereas the MoCA [31] is a more sensitive test and can detect mild cognitive impairment. However, both are recommended for use only as screening tools and not as quantitative measures of cognitive impairment.

In our previous study, instead of using dementia-oriented tests, we applied the Wechsler Adult Intelligence Scale-Fourth Edition (WAIS-IV) to test the cognitive abilities of bilaterally deafened, unilaterally implanted elderly patients before and one year after implantation [35]. According to Baddely, working memory simultaneously processes and stores information and encompasses cognitive tasks such as learning, reasoning, and language comprehension [36]. Processing speed measures the time it takes to recognize and process visual and verbal information and, secondarily, to make a decision and respond to it. Processing speed is susceptible to aging, and older individuals typically require more time to complete cognitive tasks [37]. Age-based norming, making the test comparable across age groups, is one of the significant advantages of the WAIS-IV cognitive test. One year after CI, we observed that our study group had both improved auditory parameters (speech perception and self-assessment of hearing in quiet and noise conditions and with respect to directional hearing) and cognitive abilities, represented by working memory and processing speed.

The present study aims to assess auditory and cognitive abilities, depressive symptoms, and perceived stress and their relationships in a sample of 33 patients before and after unilateral CI implantation. We used the same instruments employed in the previous report but measured outcomes not only one but also two years after implantation. In addition, the sample was larger and more diverse in terms of the laterality of hearing loss and age range.

## 2. Materials and Methods

The local Ethics Committee approved this prospective, longitudinal study (approval number: EA2/030/13, first approval 11 March 2013; amendment on 30 March 2017) conducted at a tertiary referral center. This study was conducted according to the principles of the Declaration of Helsinki. All patients signed written informed consent to participate in this study.

Thirty-three consecutive CI candidates with bilateral severe hearing loss or deafness were enrolled in this study, amounting to 14 male and 19 female patients. The mean age was 75.5 ± 4.9 years at time point one (T1) before CI.

The inclusion criteria were as follows:Age > 65 years;Indication for CI according to current German guidelines (33, 34) and meeting clinical criteria for CI surgery;Unilateral CI;German mother tongue;Complete pre- and post-implantation data collection at all time points (Figure 1).

The exclusion criteria were:Lost to follow-up appointment (due to disease and/or death);Severe visual impairment.

The tests were performed in a bright, soundproof room in the same order at each appointment. The first evaluation (T1) occurred a few months to a few days before implantation. Four weeks after implantation, auditory rehabilitation with audiological training was initiated. No additional cognitive training was included in the rehabilitation process. The second evaluation (T2) took place one year later, and the same tests were administered. Some of the T1 and T2 data on these patients were published in our previous paper [35]. The results of the third evaluation (T3), which took place two years after CI, have not been previously reported. The one-year interval between appointments was implemented to exclude possible learning effects. See Figure 1 for details on the study design.

Patients were audiologically tested before CI using pure-tone audiometry and the Freiburg Monosyllabic Speech Test (FMT) [38]. The FMT was performed during each appointment after CI. In addition, psychological status was assessed with validated questionnaires (see Table 1 for a description of the tests administered). Cognitive assessment was performed using subtests of the WAIS-IV (Wechsler Adult Intelligence Scale-Fourth Edition, Hogrefe Verlag GmbH & Co., KG, Göttingen, Germany). The WAIS-IV is a validated, standardized, and approved instrument for hearing-impaired patients with individually fitted hearing aids. In this study, the patients’ intelligence quotient (IQ) scores regarding fluid intelligence, namely, Working Memory Index (WMI) and Processing Speed Index (PSI), were evaluated and calculated based on four subtests: Digit Span and Arithmetic for WMI and Symbol Search and Coding for PSI. The PSI measures mental processing and graphomotor speed, whereas the WMI indicates concentration, attention, and working memory, formerly termed short-term memory [39,40]. The scores were scaled and converted to age-adjusted scores normalized to 100 points (with a standard deviation of 15 points), indicating average intelligence, with scores above 100 indicating above average intelligence. The WAIS-IV was scored using the original software, version 2.1.0, and age-adjusted IQ index scores were calculated.

The examiner read the test instructions, and the patients were allowed to wear hearing aids and read lips. Repetition of the instructions was permitted. The two tests, Digit Span and Arithmetic, are based on verbal instructions from the examiner; the other tests (Symbol Search and Coding) are based on visual tasks.

### Statistical Analysis

A computer-based analysis program was used to score and calculate the test results of the four applied subtests of the WAIS IV (Digit Span, Arithmetic, Symbol Search, and Coding) and to determine the index scores for WM and PS. Statistical analyses were performed using IBM SPSS Statistics 27 (IBM GmbH, Ehningen, Germany). Descriptive statistics are presented as means (MV) ± standard deviation (SD). The Kolmogorov–Smirnov test was used to test for normal distribution. Since the majority of the data were not normally distributed, non-parametric tests were used for statistical analysis. The two-tailed nonparametric Wilcoxon signed-rank test for two matched samples was used to compare the data at various time points (T1, T2, T3). The nonparametric Spearman’s rank correlation coefficient was used to measure the degree of association between two variables. The significance level was set at *p* = 0.05 (5%).

## 3. Results

### 3.1. Patient Data

The initial study group consisted of 45 patients, but 12 dropped out because of missed follow-up appointments due to illness, unwillingness to attend further study appointments, or death. Thirty-three CI candidates (nineteen women and fourteen men) were consecutively enrolled in this study, and the sample characteristics are presented in Table 2. There were several causes of the patients’ hearing loss: sudden sensorineural hearing loss (*n* = 6), noise-induced hearing loss (*n* = 4), Manière’s disease (*n* = 2), acute otitis media (*n* = 2), hereditary factors (*n* = 2), trauma (*n* = 1), and presbycusis (*n* = 1). Fifteen patients were unable to specify the causes of their hearing impairment. The patients with asymmetric hearing loss used a hearing aid on the better hearing (contralateral) ear before and after CI.

### 3.2. Hearing Abilities

Before CI, the mean Pure Tone Average (PTA) for 500 Hertz (Hz), 1000 Hz, 2000 Hz, and 4000 Hz was 92.2 ± 19.8 in the ear containing the implant (and masked contralateral ear). The FMT of the better ear was 37.76 ± 27.9% before implantation. The FMT of the worse ear designated for implantation was 5.76 ± 9.9% for word recognition (with masking of the contralateral ear). One year after CI, there was a significant improvement, increasing to 57.7 ± 20.3% (*p* = 0.005). Two years after CI, the improvement was 49.5 ± 20.8%, which was lower than that one year before (*p* = 0.044).

Subjective assessment using OI (*n* = 26) indicated significant improvement of hearing in quiet, noise, and total scores one year after CI. During the two-year follow-up, these scores were maintained at a high level. Sound localization improved significantly two years after CI. See Table 3 for the median scores with the IQR and significance levels of differences at T1, T2, and T3.

### 3.3. ADS-L

The median ADS-L scores measuring depressive symptoms were 9 (IQR 5–14) before implantation, 13.5 (IQR 6.8–20.8) at T2, and 14 (IQR 6.0–19.0) at T3. The differences between the scores were not statistically significant at any time.

### 3.4. PSQ

The mean PSQ score before CI was 0.23 (median 0.22; IQR 0.08–0.38), which is below the mean cut-off score indicating an elevated stress level (0.45). The total score remained stable at T2 and T3, and no changes in the subdomains were observed.

### 3.5. Cognition Parameters

The subtests used in this study to determine working memory (WMI) were the Digit Span and Arithmetic Tests. The processing speed (PSI) value consisted of the results for the Symbol Search and Coding Test. The results of the individual subtests are presented in Table 4. Only the arithmetic subtest score at T3 differed significantly from T1 and T2.

The median working memory index (WMI) score before CI was 95 (IQR 85.5–102.0). This median increased to 97 (IQR 89.0–107.3) after one year, but this increase was insignificant. Two years after CI, the median WMI increased significantly to 100 (IQR 89.0–106.5) (Figure 2). Twenty patients (66.0%) had an improved WMI at the two-year follow-up, nine patients (29.7%) had a lower score than that obtained preoperatively, and four patients (13.2%) had the same score.

The median PSI score improved significantly, increasing from 97 (IQR 88.0–103.0) at T1 to 103 (IQR 88.0–114.0) at T2 (*p* = 0.038). A significant improvement was also measured at T3 (median 98.5; IQR 91.0–111.0, *p* = 0.004, Figure 2). Seventeen patients (56.1%) improved their PSI scores, twelve (39.6%) scored lower, and four patients (13.2%) had a stable score two years after implantation.

### 3.6. Correlations

Correlations were calculated using Spearman’s rank correlation coefficient. No statistically significant correlations existed between working memory, processing speed, and subjective hearing ability (OI). Working memory (WMI) and years of education correlated positively at T1 (*r_s_* = 0.468, *p* = 0.01, Appendix A), T2 (*r_s_* = 0.414, *p* = 0.026, Appendix A), and T3 (*r_s_* = 0.536, *p* = 0.003, Appendix A). There was no significant correlation between years of education and processing speed. Perceived stress (PSQ) correlated positively with depressive and anxiety symptoms (ADSL) at all time points and negatively with subjective hearing ability (OI) at T2 (*r_s_* = −0.443, *p* = 0.030) and T3 (*r_s_* = −0.427, *p* = 0.030).

There was no correlation between hours of daily CI use, FMT, OI, age, or cognitive parameters. Moreover, no significant correlation between cognitive parameters and depressive symptoms (ADS-L scores) was determined at any time (Appendix A). The ADS-L scores showed a significant correlation with the PSQ (perceived stress) scores at all time points: T1 (*r_s_* = 0.74, *p* < 0.001), T2 (*r_s_* = 0.74, *p* =< 0.001), and T3 (*r_s_* = 0.772, *p* < 0.001).

## 4. Discussion

This study evaluated the neurocognitive abilities of hearing-impaired patients before and up to two years after auditory rehabilitation using CI. The neurocognitive assessment performed using the WAIS-IV test indicated a significant improvement in working memory at T3 and a significant improvement in processing speed at T2 and T3.

### 4.1. Changes in Cognitive Parameters after CI

Prior to implantation, the WAIS-IV scores for working memory and processing speed were below average compared to those for the general population (PSI: 96.77 ± 13.44; WMI: 93.45 ± 13.15; mean score: 100 ± 15). These results are consistent with the findings of many studies, such as that conducted by Lim et al. [15], in which hearing-impaired patients screened for mild cognitive impairment using the MMSE and MoCA presented significantly lower scores than the normal-hearing group. The authors concluded that there was a correlation between the hearing ability of the patients and their cognitive status.

In our present study, the PSI and WMI values improved significantly two years after CI; however, the pattern of improvement was different from that observed in our previous study [35]. Processing speed improved significantly one year after CI in both our studies. However, the previously observed significant improvement in working memory one year after CI was absent. It is very likely that the sample composition may have influenced the present results. Previously, we included only patients with bilateral deafness, whereas the present study included patients with bilateral, asymmetrical, and unilateral deafness. Nevertheless, in the present subgroup of patients with bilateral deafness, no significant improvement in working memory was observed one year after CI. The other subgroups were too small to allow us to perform a comparative between-group analysis. Including patients with different laterality of deafness may have been an important aspect responsible for the large variance seen in the WAIS-IV test results. Other factors such as the duration of deafness, cause of deafness, use of hearing aids before CI, and level of education may also have contributed to the differences seen, and this calls for more stringent patient inclusion criteria. Furthermore, the observed variance has implications for clinical practice when interviewing candidates for CI, as they may have very high hopes based on the widely publicized association between the efficacy of cochlear implantation and cognitive improvement.

Corroborating our study, Jorgensen et al. [29] showed that hearing impairment negatively affected MMSE scores in adolescents. A similar conclusion was reached by Dupuis et al. [44], who examined the effects of hearing and vision impairment on MoCA scores and suggested that cognitive impairment was over-diagnosed among patients with sensory deficits. This issue was addressed by Claes et al. [32,45,46] and Calvino et al. [47] by testing CI recipients with the Repeatable Battery for the Assessment of Neuropsychological Status for Hearing Impaired individuals (RBANS-H), which is a modified version of the RBANS neurocognitive test with additional written instructions to rule out perceptual difficulties due to orally presented instructions. Claes et al. [46] showed that the total score and the subdomains of immediate and delayed memory and attention improved significantly 12 months after CI, supporting our earlier results and, partially, our present results [35].

Völter et al. [33] developed a computerized non-auditory test battery called ALAcog for the neurocognitive assessment of CI patients. The ALAcog consists of eight subtests assessing different neurocognitive abilities (attention, short-term, and long-term memory; working memory; processing speed; and executive functions). Using ALAcog, they found that the hearing-impaired adults performed significantly worse than the age-matched controls in some neurocognitive subtests. Furthermore, they found a significant cognitive improvement among bilaterally hearing-impaired patients over 50 twelve months and up to 65 months after implantation [48], suggesting a sustained cognitive benefit in this population, further supporting our findings.

### 4.2. Correlations

The present study found a correlation between years of education and working memory at all time points (T1–T3), suggesting that a longer period of education positively affects working memory before and after implantation. Such a correlation has also been found in a group of healthy older adults, both functionally (greater working memory-related activity in the left prefrontal cortex) and structurally (larger right-medial frontal and middle cingulate gyri and right inferior parietal lobules in those with more years of education) [49], thus supporting the general significance of this finding. Our findings are also supported by the work conducted by Calvino et al., who demonstrated a positive correlation between RBANS-H and years of education [50], and by Völter et al., who showed a correlation between the years of education and the improvement in patients’ cognitive ability after CI [51]. Therefore, our present results strengthen the evidence for an association between years of education and the neurocognitive parameter working memory in the context of auditory rehabilitation.

In addition, a significant correlation between working memory and processing speed was found one and two years after CI but not before CI. A recent study examining associations between cognitive domains, such as working memory and processing speed, and successful aging found a strong relationship between WM, PS, and a domain of successful aging defined as the absence of disease [52]. Although deafness or hearing impairment were not on the list of diseases included in this research, it is tempting to speculate that individuals affected by deafness may perceive this condition as a major disease. Consequently, after effective auditory rehabilitation, the level of successful aging increases, as reflected by the restored relationship between WM and PS.

There was no statistically significant correlation between cognitive parameters and speech perception (FMT and OI) or depressive symptoms (ADS-L).

### 4.3. Strengths, Limitations, and Perspectives

The major strength of this study is its use of neurocognitive testing at multiple time points: pre-implantation, one year, and two years post-implantation. Thus, a learning effect can be excluded due to the large interval between assessments (one year). Another advantage of this study is that neurocognitive testing was conducted using the WAIS-IV, a qualitative and quantitative measure of WMI and PSI and therefore a more sensitive test than others used to screen for cognitive impairment. In addition, the WAIS-IV is an age-adjusted instrument, so it can be used to assess all adolescents and adults older than 16 years, and the scores can be compared across age groups.

A limitation of this study is its relatively small sample size, especially with respect to the subgroups. In addition, the heterogeneous duration and causes of hearing loss, as well as the variable use of hearing aids before cochlear implantation, might have influenced the results of neurocognitive testing. Future studies should aim to include more patients and allow for stratification according to age, gender, and education level, as well as type, duration, and degree of hearing loss. Alternatively, a homogeneous group of patients could be recruited, primarily if the research question is sharply focused, for example, to compare patients with the same type of hearing loss but with two different durations (e.g., short and long). The second limitation of our study is that only one test was used to assess cognitive abilities. Using an additional tool, preferably a non-verbal one (such as ALAcog), would benefit this type of study.

It remains unclear whether cognitive ability level correlates with the duration of hearing loss and whether there are other associated factors, such as type of professional occupation, hours of communication per day, or frequency and duration of social activities. Thus, it would be interesting to learn more about these possible confounding factors in future studies.

## 5. Conclusions

The significant improvement in working memory and processing speed during the 2-year follow-up after CI suggests that auditory rehabilitation with CI has a positive effect on the cognitive abilities of the patients receiving implants. The significant improvement in directional hearing and working memory observed 2 years after CI despite a decrease in FMT scores at this time point indicates a cognitive improvement in implanted patients that was delayed and not directly correlated with word recognition scores. Further studies with well-defined samples should further evaluate the effect of age, hearing impairment, and auditory training on patients’ cognitive abilities.

## Figures and Tables

**Figure 1 brainsci-13-01673-f001:**
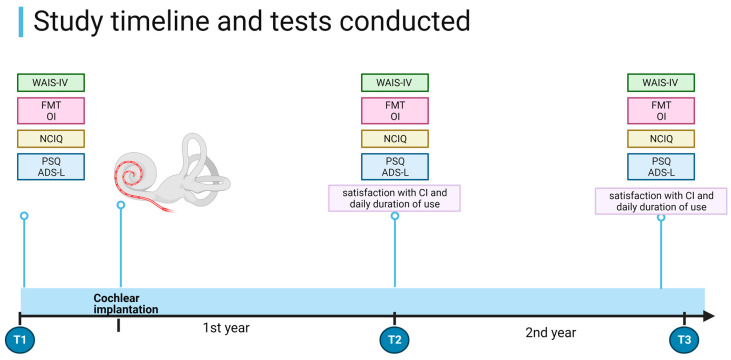
Study timeline and procedures. WAIS-IV = Wechsler Adult Intelligence Scale-Fourth Edition. FMT = Freiburg Monosyllable Test; OI—Oldenburg Inventory; PSQ—Perceived Stress Questionnaire; ADS-L—General Depression Scale. T1 = before CI, T2 = 1 year after CI, and T3 = 2 years after CI. Created using RioRender.com.

**Figure 2 brainsci-13-01673-f002:**
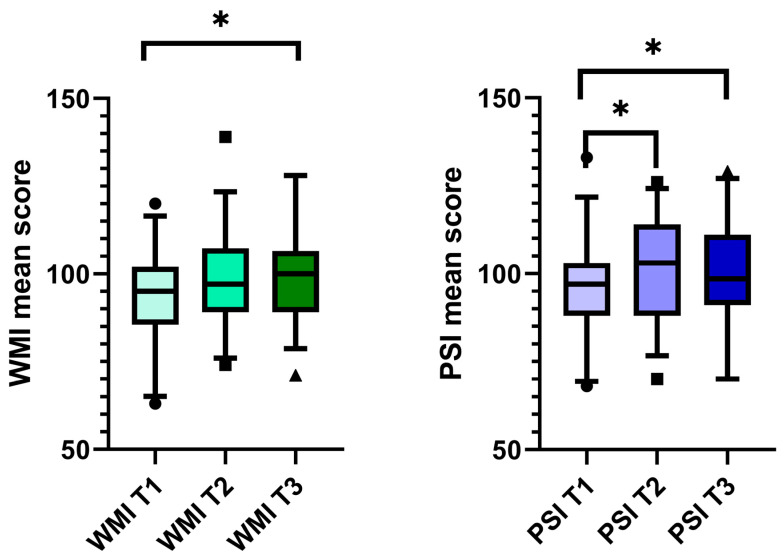
Working memory and processing speed improved after CI. Boxplot with the upper and lower whiskers, first and third quartiles, and median scores of WMI and PSI. The Wilcoxon signed-rank test was used to compare the data at time points T1–T3. Only the significantly different comparisons were marked. WM I = working memory index; PSI = processing speed index; T1 = before CI, T2 = 1 year after CI, and T3 = 2 years after CI; * *p* < 0.05.

**Table 1 brainsci-13-01673-t001:** Questionnaires applied in this study.

Oldenburg Inventory (OI)	The OI [41] assesses a patient’s subjective hearing ability using questions categorized into 3 subdomains, namely, “hearing in quiet”, “hearing with background noise”, and “localization”, and an overall score.
Perceived Stress Questionnaire (PSQ)	The PSQ is used to assess perceived stress levels [42]. It consists of 30 items measuring four subscales: worries, tension, joy, and demands. The cut-off score for low-degree stress is 0.45.
General Depression Scale (ADS-L)	The General Depression Scale (ADS) [43] assesses depressive symptoms. The test consists of 20 items with scores ranging from 0 to 60, with a cut-off score of 23. In this study, the long version (ADS-L) was used.

**Table 2 brainsci-13-01673-t002:** Patient characteristics. SD = standard deviation; AHL = asymmetric hearing loss; DSD = double-sided deafness (bilateral deafness); SSD = single-sided deafness (unilateral deafness).

The total number of patients included	33
Sex	female = 19; male = 14
Age (mean ± SD, range)	75.5 ± 4.9; 65–88
Years of education (mean ± SD, range)	12.76 ± 2.36; 8–18
Laterality of hearing loss Duration of deafness (mean ± SD, range)	AHL (*n* = 10); DSD (*n* = 22 *); SSD (*n* = 1) 12.8 ± 13.4; 0.5–58

* indicates that thirteen DSD patients from this subgroup were included in the previous study, where the results were measured one year after CI. The two-year results for these patients have not yet been reported.

**Table 3 brainsci-13-01673-t003:** Oldenburg inventory (OI) scores before CI (T1) and at 1 year (T2) or 2 years after implantation (T3). The Wilcoxon signed-rank test was used to compare the data at the time points T1–T2, T1–T3, and T2–T3. OI = Oldenburg inventory. IQR = interquartile range. The asterisk (*) indicates significance.

	T1		T2		T3		Level of Significance for T1–T2	Level of Significance for T1–T3	Level of Significance for T2–T3
	Median	IQR	Median	IQR	Median	IQR	Two-Tailed	Two-Tailed	Two-Tailed
OI-quiet	2.6	2.2–3.4	3.4	2.8–4.3	3.6	2.5–4.1	*p* = 0.003 *	*p* = 0.002 *	*p* = 0.572
OI-noise	2	1.5–2.2	2.5	2.2–3.0	2.4	1.8–3.1	*p* < 0.001 *	*p* < 0.001 *	*p* = 0.224
OI-localization	2.5	2.0–3.0	3	2.0–3.5	3	2.0–4.0	*p* = 0.051	*p* = 0.025 *	*p* = 0.329
OI-total	2.3	1.9–2.6	3	2.4–3.4	2.96	2.2–3.4	*p* < 0.001 *	*p* = 0.001 *	*p* = 0.526

**Table 4 brainsci-13-01673-t004:** Scores for the cognition parameters (working memory index and processing speed index) and the subtests performed in this study. The Wilcoxon signed-rank test was used to compare the data at time points T1–T2, T1–T3, and T2–T3. The asterisk (*) indicates significance.

	T1		T2		T3		Level of Significance for T1–T2	Level of Significance for T1–T3	Level of Significance for T2–T3
	Median	IQR	Median	IQR	Median	IQR	Two-Tailed	Two-Tailed	Two–Tailed
Digit span	22	19–26	22.5	20.3–23	23	20.0–26.5	*p* = 0.794	*p* = 0.905	*p* = 0.769
Arithmetic	13	11–17	14	12–16.0	14	13.0–17.5	*p* = 0.215	*p* = 0.001 *	*p* = 0.009 *
Symbol Search	20	14.5–24.0	22	16.0–26.0	21.0	18.0–26.0	*p* = 0.178	*p* = 0.557	*p* = 0.647
Coding	49	38.5–56.0	50	36.0–61.0	50	37.5–58.0	*p* = 0.206	*p* = 0.250	*p* = 0.918

IQR = interquartile range.

## Data Availability

The data presented in this study are available on request from the corresponding author. The data are not publicly available due to privacy.

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
