# Peer review of "Sustained Cognitive Improvement in Patients over 65 Two Years after Cochlear Implantation"

_brainsci, 2023, doi:10.3390/brainsci13121673_

Round 1
Reviewer 1 Report
Comments and Suggestions for Authors
The manuscript reports auditory and cognitive abilities, depressive symptoms, and perceived stress of a group of CI users older than 65, two years after cochlear implantation. The issue is interesting but in my opinion the manuscript is too long and various argument should be synthetized.
Some minor issues:
lines 138, 139, 140: "The initial study group consisted of 45 patients, but 12 dropped out because of missed follow-up appointments due to illness, unwillingness to attend further study appointments or death" should be reported in the results chapter
when non parametric statistical tests are used, data should be reported as median and IQR
Check the references: for example reference 1 and 2 are wrong
Author Response
The manuscript reports auditory and cognitive abilities, depressive symptoms, and perceived stress of a group of CI users older than 65, two years after cochlear implantation. The issue is interesting but in my opinion the manuscript is too long and various argument should be synthetized.
The Reviewer’s comments are appreciated. In response to this constructive criticism, we shortened and revised our manuscript to make it concise. We rewrote the Introduction and Discussion and introduced indicated changes in the Materials & Methods, and Results.
Some minor issues:
lines 138, 139, 140: "The initial study group consisted of 45 patients, but 12 dropped out because of missed follow-up appointments due to illness, unwillingness to attend further study appointments or death" should be reported in the results chapter
We followed this suggestion and moved this information to the Results.
when non parametric statistical tests are used, data should be reported as median and IQR
The comments are appreciated – we changed the reporting accordingly and used median and IQR for data reporting.
Check the references: for example reference 1 and 2 are wrong
We have revised the Introduction and Discussion and double-checked the accuracy of the literature cited.
Reviewer 2 Report
Comments and Suggestions for Authors
Thank you for submitting this manuscript. A few comments on how to improve it:
1. Consider including in your introduction the Livingston et al. 2020 Lancet Report on Dementia and the review by Yeo et al 2023. They are both very important articles in the field.
2. Consider moving the heatmap to an appendix as it too big/complex for the main body
3. Figure 2 is your main finding so please make it easier to follow and a bit more self-explanatory
4. Even though you report statistically significant effects, the variance in the WMI and PSI scores is so high that it is very difficult to see any meaningful difference in the boxplots. You need to at least discuss/justify this as it is your main finding...
5. It is unclear if this study reports results from the previous one with some additional ones for Year 2 or if it is a brand new study. I guess the former which is ok but you need to make this clear.
6. You need to report a lot more about your participants' hearing loss and CI, e.g. years of deafness, age at implantation, use of hearing aids etc. There are so many factors that could influence your results and you need to explore at least some of them or at least discuss them. You could at least explore correlations with age which is the simplest. The variance in the scores that I mentioned above is related to this.
Author Response
Thank you for submitting this manuscript. A few comments on how to improve it:
1. Consider including in your introduction the Livingston et al. 2020 Lancet Report on Dementia and the review by Yeo et al 2023. They are both very important articles in the field.
We are grateful for the suggestion and now cite the recommended references as follows:
“The aging process is associated not only with hearing loss but also with cognitive decline. Several studies focusing on dementia and hearing impairment provided evidence suggesting that the degree of hearing impairment is related to cognitive decline [13-15]. Livingston and colleagues proposed that hearing loss is the most important, modifiable risk factor for age-related dementia in middle-aged persons between 45 and 65 [16,17] and that its treatment could reduce the risk of dementia by 8.2% [17].
Hearing rehabilitation with hearing aids for patients with moderate to severe hearing loss was associated with improved cognition [18,19] and suggested to counteract cognitive decline [20,21]. In particular, older candidates with severe to profound hearing loss appear to benefit from hearing rehabilitation with CI. A recent systematic review and meta-analysis by Yeo and colleagues demonstrated a decrease of 19% in cognitive decline among hearing-impaired users of hearing aids and cochlear implants [22]. Moreover, in the same work, the authors observed a 3% improvement in cognitive test scores of that population.”
- Consider moving the heatmap to an appendix as it too big/complex for the main body
The comment is appreciated – we moved the heatmap to supplementary information.
- Figure 2 is your main finding so please make it easier to follow and a bit more self-explanatory
We edited Figure 2 and removed the non-significance from the plot to clarify the message.
- Even though you report statistically significant effects, the variance in the WMI and PSI scores is so high that it is very difficult to see any meaningful difference in the boxplots. You need to at least discuss/justify this as it is your main finding...
The comments are appreciated. We addressed them in the Discussion in the following way:
“Including patients with different laterality of deafness may have been an important aspect responsible for the large variance seen in the WAIS-IV test results. Other factors such as duration of deafness, cause of deafness, use of hearing aids before CI, and level of education may also have contributed to the differences seen, which calls for more stringent patient inclusion criteria. Furthermore, the observed variance has implications for clinical practice when interviewing candidates for CI, as they may have very high hopes based on the widely publicized association between the efficacy of cochlear implantation and cognitive improvement.”
- It is unclear if this study reports results from the previous one with some additional ones for Year 2 or if it is a brand new study. I guess the former which is ok but you need to make this clear.
We addressed this comment in MM and in Table 2:
“Some of the T1 and T2 data of these patients' were published in our previous paper [35]. The results of the third evaluation (T3), which took place two years after CI, have not been previously reported.”
“13 DSD patients from this subgroup were included in the previous study, where the results were measured one year after CI. The two-year results for these patients have not yet been reported.”
- You need to report a lot more about your participants' hearing loss and CI, e.g. years of deafness, age at implantation, use of hearing aids etc. There are so many factors that could influence your results and you need to explore at least some of them or at least discuss them. You could at least explore correlations with age which is the simplest. The variance in the scores that I mentioned above is related to this.
We have added the requested information:
“There were several causes for the patient´s hearing loss: sudden sensorineural hearing loss (n=6), noise-induced hearing loss (n=4), Manière’s disease (n=2), acute otitis media (n=2), hereditary (n=2), trauma (n=1) and presbyacusis (n=1). Fifteen patients were unable to specify the causes of their hearing impairment. Patients with asymmetric hearing loss used a hearing aid on the better hearing (contralateral) ear before and after CI.”
We also added information about the deafness duration.
There was no correlation between age and cognitive scores at any time. We report it now in the Results.
Round 2
Reviewer 1 Report
Comments and Suggestions for Authors
The manuscript is still too long
Reviewer 2 Report
Comments and Suggestions for Authors
Thank you for addressing my comments